# Rice Cultivation without Synthetic Fertilizers and Performance of Microbial Fuel Cells (MFCs) under Continuous Irrigation with Treated Wastewater

**Dong Duy Pham** [1,*]**, Kei Cai** [2]**, Luc Duc Phung** [3]**, Nobuo Kaku** [2]**, Atsushi Sasaki** [4]**,
Yuka Sasaki** [2]**, Kenichi Horiguchi** [2]**, Dung Viet Pham** [2] **and Toru Watanabe** [2]

1  Faculty of Environmental Engineering, National University of Civil Engineering, 55 Giai Phong Street,
   Hai Ba Trung District, Hanoi 100803, Vietnam
2  Faculty of Agriculture, Yamagata University, 1-23 Wakaba-machi, Tsuruoka, Yamagata 997-8555, Japan
3  The United Graduate School of Agricultural Sciences, Iwate University, 3-18-8 Ueda, Morioka,
   Iwate 020-8550, Japan
4  Faculty of Engineering, Yamagata University, 4-3-16 Jonan, Yonezawa, Yamagata 992-8510, Japan
*  Correspondence: dongpd@nuce.edu.vn

**Abstract:** To obtain a high rice yield and quality for animal feed without synthetic fertilizers, an experiment with bench-scale apparatus was conducted by applying continuous irrigation with treated municipal wastewater (TWW). Uniform rice seedlings of a high-yield variety (*Oryza sativa* L., cv. Bekoaoba) were transplanted in five treatments to examine different TWW irrigation directions ("bottom-to-top" and "top-to-top" irrigation) and fertilization practices (with and without P-synthetic fertilizers) as well as one control that simulated the irrigation and fertilization management of normal paddy fields. The highest rice yield (14.1 t ha$^{-1}$), shoot dry mass (12.9 t ha$^{-1}$), and protein content in brown rice (14.6%) were achieved using bottom-to-top irrigation, although synthetic fertilizers were not applied. In addition, this subsurface irrigation system could contribute to environmental protection by removing 85–90% of nitrogen from TWW more effectively than the top-to-top irrigation, which showed a removal efficiency of approximately 63%. No accumulation of heavy metals (Fe, Mn, Cu, Zn, Cd, Ni, Pb, Cr, and As) in the paddy soils was observed after TWW irrigation for five months, and the contents of these metals in the harvested brown rice were lower than the permissible limits recommended by international standards. A microbial fuel cell system (MFC) was installed in the cultivation system using graphite-felt electrodes to test the capacity of electricity generation; however, the electricity output was much lower than that reported in normal paddy fields. Bottom-to-top irrigation with TWW can be considered a potential practice to meet both water and nutrient demand for rice cultivation in order to achieve a very high yield and nutritional quality of cultivated rice without necessitating the application of synthetic fertilizers.

**Keywords:** continuous irrigation; bottom-to-top irrigation; nutritional quality of rice; rice for animal feeding; synthetic fertilizers; microbial fuel cell (MFC)

## 1. Introduction

Shortages of irrigation water along with reinforced pressures on regional available water resources owing to population growth, urbanization, industrialization, and climate change [1] are an important driver in the reuse of treated municipal wastewater (TWW) for agricultural activities, especially in the irrigation of rice (*Oryza sativa* L.) paddy fields. The reuse of TWW has been demonstrated to diminish water scarcity, alleviate the degradation of wastewater-receiving environments, and conserve other freshwater sources [2]. In addition, irrigation with TWW is advantageous to the improvement

of soil fertility and rice productivity while reducing the use of commercial fertilizers [3] as a result of the substantial amounts of plant nutrients such as N, P, K, and organic matter available in TWW. However, these benefits are likely countered by substantial concerns about human health risks, owing to potential pathogenic microorganisms, heavy metals, and other contaminants in wastewater [4,5]. Thus, it is necessary to develop new irrigation management strategies and technologies to optimize the advantages while minimizing the aforementioned downsides.

We recently introduced an innovative rice cultivation system in which TWW was continuously supplied into paddy fields throughout the crop seasons in order to achieve very high yields and superior rice protein content of a forage rice without the supplementation of N fertilizers. However, P fertilizer was still applied at a high rate (160 kg $P_2O_5$ ha$^{-1}$) to paddies in the system [6]. As one of the most important nutrients for agricultural production, P demand is always increasing. This has caused considerable pressure on the mining of phosphate rock, which is projected to be significantly depleted by the end of this century [7]. Furthermore, P utilization efficiency is particularly low, owing to significant losses [8] that have led to serious environmental problems such as red tide and eutrophication [7].

Reducing the use of P fertilizers and enhancing use efficiency are not only critical for alleviating the negative impacts of P on the natural environment but also probably conducive to farmers achieving a higher profit. On the other hand, at the end of the previous crop season [6], a significant increase of P content in the paddy soils irrigated with TWW and simultaneously supplemented with P fertilizer raised a hypothesis that the high yields and superior rice protein content might be maintained during subsequent crop seasons without necessitating the supplementation of synthetic P fertilizers. However, the rate of soil P buildup may slow down and eventually reach steady-state after long-term TWW irrigation [9] due to the downward movement of P binding minerals. In such case, there is the possibility that P input with TWW and steady-state soil P levels are insufficient to meet the P demand, so P fertilization might again become necessary [6].

In addition to water scarcity, an increasing energy demand has been an important issue facing the world nowadays [10]. This has triggered an ongoing motivation to find and develop renewable energy sources. Among possible alternatives for renewable bioenergy, the application of microbial fuel cells (MFCs) has been suggested as a promising approach to generate electricity directly by using bacteria to break down organic substrates under the anaerobic condition of paddy soil [11,12]. As a result, considerable effort has been made to evaluate and advance the performance of MFCs associated with paddy fields [11,13–16].

Apart from the essential plant nutrients (N, P, K) and micronutrients (Na, Ca, Mg, etc.), TWW also contains a significant amount of organic matter (OM), which might benefit the electricity generation of MFCs in paddy fields irrigated with TWW [17]. Thus, we hypothesized that the performance of MFCs could be effectively elevated in the fields employing continuous irrigation with TWW by supplying substantial amounts of OM into the paddy soil. Though we attempted to investigate the performance of MFCs under such irrigation methods, the electricity generation was not assessed properly, owing to a number of unexpected failures in electrode connections and the operation of the MFCs during our previous study [6]. In this follow-up experiment, we modified the MFC systems to avoid such failures and anticipated that the electricity generation could be thoroughly estimated in response to continuous irrigation with TWW.

Overall, the present study aims to test the aforementioned hypotheses. The specific objective of the study is to investigate (1) the effects of continuous irrigation with TWW on the performance of rice plants and changes in soil nutrition with and without the supplementation of P fertilizers, and (2) the performance of MFCs in paddy fields under different conditions of continuous irrigation with TWW.

## 2. Materials and Methods

### 2.1. Experimental Design

An experiment was conducted using the same bench-scale apparatus that was used in our previous study [6], which simulated paddy fields of 0.18 m$^2$ (0.3 m × 0.6 m) (Figure 1) in order to examine two directions of continuous irrigation with TWW: Bottom-to-top irrigation (BI) and top-to-top irrigation (TI). Two fertilization managements were used: The fertilization of 160 kg $P_2O_5$ ha$^{-1}$ as basal (F$_P$) and no exogenous chemical fertilization (F$_0$).

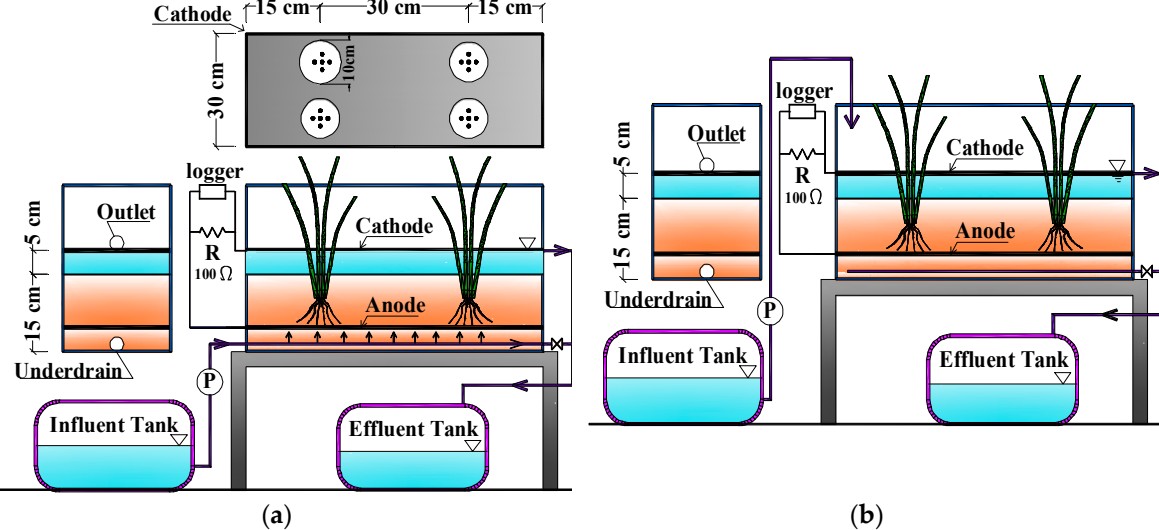

**Figure 1.** Simulated paddy fields with different directions of continuous treated municipal wastewater (TWW) irrigation: (**a**) Bottom-to-top irrigation; and (**b**) top-to-top irrigation.

Continuous irrigation was carried out by pumping TWW that was collected from a local wastewater treatment plant in Tsuruoka city, Japan, from an influent tank into the containers, either through underground pipes installed at the bottoms of the containers (BI) or on the soil surface (TI) at a constant flow rate of 4.5 L day$^{-1}$. The underground pipes were covered with a thin layer of gravel and filled to 15 cm from the bottom using paddy soil. Effluents from the experimental containers were collected in different effluent tanks via overflow pipes fixed at a height of 5 cm above the soil surface in all containers (Figure 1). The cultivation was implemented either by using a new 20 cm soil layer of a local paddy field (New) or reusing the soils previously irrigated with TWW, as in [6] (Old).

Overall, as shown in Table 1, there were five treatments (A, B, C, E, and F) combining the examined factors. Treatments B and E were assigned to apply BI at a constant flow rate of 4.5 L day$^{-1}$, since this setup was the most favorable irrigation management producing the highest grain yield and rice protein content in our previous study [6]. A commercial P-fertilizer (14% P) was applied in Treatment E at 20.6 g for 0.18 m$^2$ equivalent to 160 kg P ha$^{-1}$ as basal (Fp), while no synthetic fertilizer was supplied into B (F$_0$). Both treatments reused the soil previously irrigated with TWW and supplemented with a P-fertilizer (old) in the precursor study [6]. Additionally, the circuit of MFCs were connected in both treatments (closed). Treatment C was a setup exactly the same as Treatment B, except that the circuit of its MFC was disconnected (opened) in order to inspect the impacts of operating MFCs on the growth and development of the rice plants.

In attempt to elaborate the performance of MFCs under the different methods of continuous irrigation with TWW, Treatment A and Treatment F were assigned to employ BI and TI using new soil (new), respectively. The TWW was also supplied at a flow rate of 4.5 L day$^{-1}$, as implemented in the previous treatments. Treatment D (control) illustrated the cultivation conditions of a normal paddy field where irrigation was manually carried out day-by-day using tap water (TW) to compensate for

the water loss due to evapotranspiration. The fertilization for Treatment D was accomplished by applying a commercial N–P–K fertilizer (20.6 g 14-14-14 N–P–K fertilizer for 0.18 m$^2$, equivalent to 160 kg N ha$^{-1}$, 160 kg P ha$^{-1}$, and 160 kg K ha$^{-1}$) before transplanting and an N–K fertilizer (9.0 g 14-14 N–K fertilizer for 0.18 m$^2$ equivalent to 100 kg N ha$^{-1}$ and 100 kg K ha$^{-1}$) as a top dressing at the panicle initiation stage (F$_{NPK}$).

**Table 1.** Cultivation conditions of experimental treatments.

| Cultivation Conditions | Experimental Treatments | | | | | |
|---|---|---|---|---|---|---|
| | **Treatment A** | **Treatment B** | **Treatment C** | **Treatment D** | **Treatment E** | **Treatment F** |
| **Soil type** | New | Old | Old | New | Old | New |
| **Water type** | TWW | TWW | TWW | TW | TWW | TWW |
| **Flow rate (L day$^{-1}$)** | 4.5 | 4.5 | 4.5 | Daily manual watering to compensate for ET | 4.5 | 4.5 |
| **Flow direction** | BI | BI | BI | | BI | TI |
| **Irrigation regimes** | Continuous | Continuous | Continuous | | Continuous | Continuous |
| **Fertilization** | F$_0$ | F$_0$ | F$_0$ | F$_{NPK}$ | F$_P$ | F$_0$ |
| **MFC circuit** | Closed | Closed | Opened | Closed | Closed | Closed |

Old soil: Soil was reused from 2015 season [6]; TW: Tap water; TWW: Treated municipal wastewater; new soil: Soil was taken from surface layer of Yamagata University farm; BI: Bottom-to-top irrigation; TI: Top-to-top irrigation; F$_0$: No fertilization; F$_P$: P$_2$O$_5$ Fertilizer application; F$_{NPK}$: N–P–K fertilizer application.

## 2.2. Crop and Water Management

After soil puddling, uniform 30-day-old seedlings of a forage rice variety (*Oryza sativa* L., cv. Bekoaoba) were transplanted on 20 May 2016, at a rate of five seedlings per plant and four plants per container, as illustrated in Figure 1. During the growing period, from transplanting to harvesting on 28 September 2016, a water depth of 5 cm was maintained in treatments A, B, C, E, and F by the pumping systems. Irrigation was intermittent only from 4 July to 11 July—known as midseason drainage (MSD)—when the water supply was stopped and the paddy soils were drained to enhance rice root growth by exposing the rice root zones to more oxygen.

## 2.3. Crop Growth and Yield Monitoring

During the vegetative phase, leaf chlorophyll meter readings were recorded using the soil plant analysis development (SPAD) method with a chlorophyll meter (SPAD-502) [18], while the plant height (cm) was measured from the ground level to the topmost panicle. At harvesting, yield components such as the number of panicles per plant, number of spikelets per panicle, filled spikelet percentage, and 1000-grain weight (g) were determined. The grain yield (g), measured as the weight of brown rice, was adjusted to 15% moisture content. The shoot dry matter (g) was determined by measuring the weight of rice shoots dried at 80 °C for 48 h.

## 2.4. Quality of Brown Rice

The quality of brown rice was evaluated according to AOAC [19] by estimating five nutritional components: Crude protein, fat, fiber, nitrogen-free extract (NFE), and organic matter (OM). The concentrations of total nitrogen (TN), total phosphorous (TP), and heavy metals were estimated using the respective standard methods [20].

## 2.5. Water and Soil Monitoring

The quality of the irrigation TWW was monitored on a weekly basis. Water samples were analyzed following the methods described by [6] for basic parameters including TN, TP, pH, electrical conductivity (EC), and heavy metal concentrations. The monthly average values of the investigated characteristics of irrigation TWW over the crop growth season are listed in Table 2. Soil samples before

and after the experiment were collected and analyzed for the content of elements including N, P, K (Table 3), and common heavy metals were collected by following standard methods [20].

**Table 2.** Monthly average water quality of influent TWW.

| Parameters | Units | Months | | | | | Average | Permissible Limits * |
| | | May | Jun. | Jul. | Aug. | Sep. | | |
|---|---|---|---|---|---|---|---|---|
| pH | - | 7.3 | 7.0 | 7.3 | 7.2 | 7.0 | 7.1 | NA |
| EC | mS m$^{-1}$ | 60.8 | 72.5 | 63.5 | 64.5 | 56.7 | 63.6 | NA |
| TN | mg L$^{-1}$ | 28.8 | 40.0 | 26.8 | 28.6 | 23.0 | 27.6 | NA |
| TP | mg L$^{-1}$ | 0.23 | 0.16 | 0.15 | 0.15 | 0.18 | 0.17 | NA |
| K | mg L$^{-1}$ | 11.4 | 13.0 | 13.2 | 14.4 | 13.0 | 13.0 | NA |
| Cu | µg L$^{-1}$ | 15.4 | 11.0 | 10.8 | 8.4 | 10.6 | 11.2 | 200 |
| Cr | µg L$^{-1}$ | 0.60 | 0.60 | 0.60 | 0.60 | 0.80 | 0.64 | 10 |
| Zn | µg L$^{-1}$ | 58.0 | 50.4 | 45.0 | 44.0 | 40.2 | 47.5 | 2000 |
| Cd | µg L$^{-1}$ | NA | NA | NA | NA | NA | NA | 10 |
| Pb | µg L$^{-1}$ | 0.80 | 0.60 | 0.80 | 0.60 | 0.60 | 0.68 | 5000 |
| As | µg L$^{-1}$ | NA | NA | NA | NA | NA | NA | 100 |
| Mn | µg L$^{-1}$ | 42.2 | 24.0 | 32.8 | 24.4 | 32.4 | 31.1 | 200 |
| Fe | µg L$^{-1}$ | 88.2 | 78.8 | 99.8 | 100.4 | 151.8 | 103.8 | 5000 |
| Ni | µg L$^{-1}$ | 7.4 | 18.2 | 11.8 | 11.4 | 10.0 | 11.8 | 200 |

*: Adapted from [21]; NA: Not available.

**Table 3.** Nutrients (kg ha$^{-1}$) in experimental soil before and after experiment.

| Treatment | TN | | TP | | TK | |
| | Before Experiment | After Experiment | Before Experiment | After Experiment | Before Experiment | After Experiment |
|---|---|---|---|---|---|---|
| A | 1439 | 1337 | 567 | 534 | 7775 | 4998 |
| B | 1599 | 1445 | 899 | 853 | 6760 | 8553 |
| C | 1599 | 1547 | 899 | 833 | 6760 | 7320 |
| D | 1439 | 1520 | 567 | 642 | 7775 | 6599 |
| E | 1599 | 1373 | 899 | 946 | 6760 | 7007 |
| F | 1439 | 1474 | 567 | 548 | 7775 | 6272 |

## 2.6. Electricity Generation Monitoring

In all treatments, the systems of MFCs were configured fundamentally in the same way as that described by [6] with minor adjustments. Two rectangular (0.3 m × 0.6 m) graphite electrodes were implemented to construct an MFC system (Figure 1), in which an anode was inserted in the soil approximately 10 cm below the soil surface and a cathode was placed right above the water surface. The distance between the electrodes was 15 cm. In order to allow for rice transplantation and growth, four holes (diameter = 10 cm) were punched in the cathode. In all treatments, the electrodes were connected to a 100-Ω external resistor (closed MFC circuit), across which the voltage was monitored automatically every 10 min using a data logger (GL220, Graphtec, Japan)—except Treatment C, in which the electrodes were not connected (open MFC circuit). To avoid the copper cable oxidation seen in the previous study [6], graphite rods were used to connect the two electrodes.

## 2.7. Statistical Analysis

The treatments were not replicated owing to resource limitations. Instead, four plants of rice seedlings were transplanted into each treatment and were considered as replicates during data analysis. The data collected from the experiment were subjected to an analysis of variance (ANOVA), and the means for significant treatment effects were compared using Tukey's honestly significant difference (HSD) at a 5% probability level using the statistical software package SPSS 24.0 (IBM Corp. Released 2016. IBM SPSS Statistics for Windows, Version 24.0. IBM Corp., Armonk, NY, USA).

## 3. Results and Discussion

### 3.1. Irrigation Water Quality and Nitrogen Removal Efficiency

Table 2 shows the chemical characteristics of TWW that were examined monthly during the experimental period. The pH of the irrigation TWW was maintained between 7.0 and 7.3. The EC varied from 56.7 to 72.5 mS m$^{-1}$. The average TN varied from 23.0 to 31.0 mg L$^{-1}$, K from 11.4 to 14.4 mg L$^{-1}$, and TP from 0.15 to 0.23 mg L$^{-1}$. The concentrations of heavy metals in TWW satisfied the recommended maximum concentrations of trace elements in irrigation water [21], despite the different variations during the five-month growing period.

The monitoring of TN concentration in the effluent tanks revealed that the average N removal efficiencies in treatments A, B, C, and E were 85%, 90%, 86%, and 86%, respectively (Figure 2). The removal efficiency obtained from Treatment F (63%) was much lower, implying that N removal was enhanced by the infiltration of TWW through the paddy soil layers under BI. This observation was consistent with our previous finding [6].

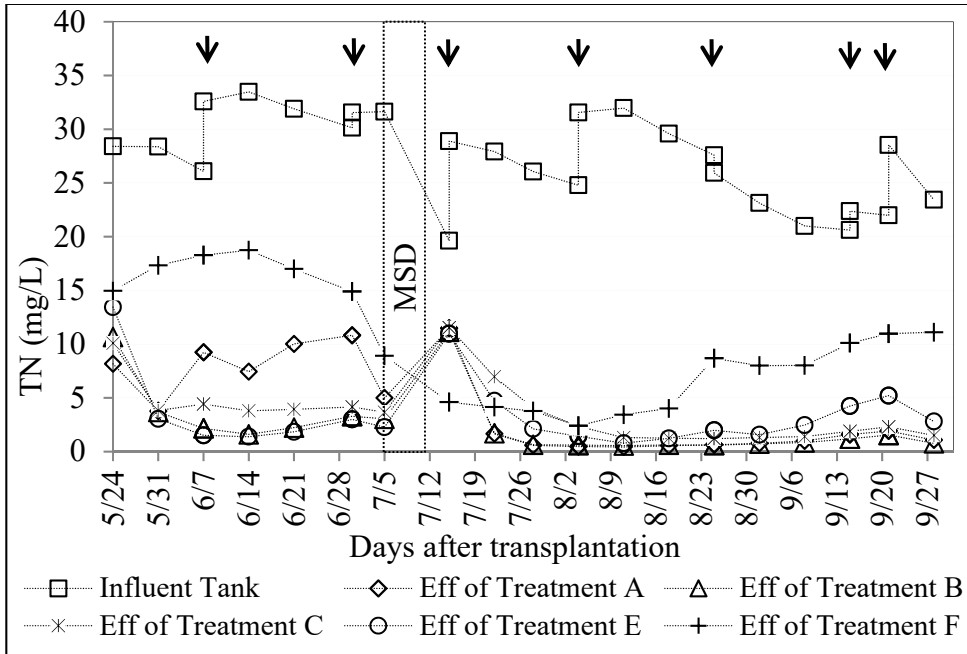

**Figure 2.** Total nitrogen (TN) of the irrigation water. Solid arrows indicate dates when treated wastewater was added to the influent tank, and MSD means midsummer drainage to dry up the soil layers.

The fates of nitrogen removed from TWW are illustrated in Figure 3. In all treatments, the largest part of removed nitrogen was absorbed in to rice plants, followed by those which were emitted into the atmosphere.

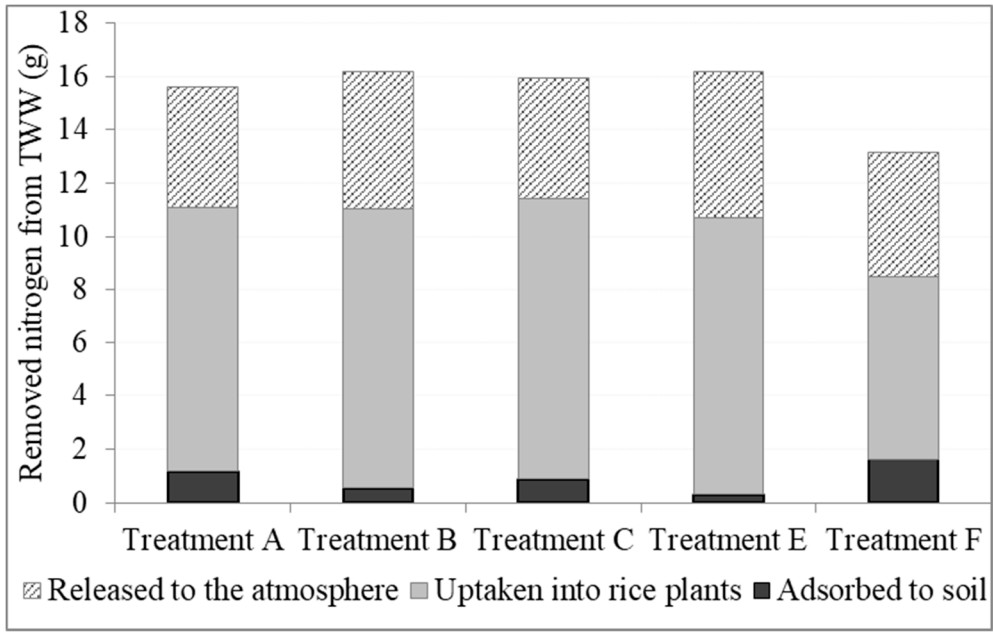

**Figure 3.** Fate of nitrogen removed from irrigation water.

*3.2. Crop Growth*

The growth parameters of rice, such as plant height, SPAD values, and shoot dry mass, are shown in Table 4. BI promoted the better growth of rice plants in treatments A, B, C, and E, with plant heights of above 101 cm. Meanwhile, TI (Treatment F) and the control water management (Treatment D) could only produce plants with relatively shorter heights of less than 100 cm (Table 4). The same range of plant heights in Treatments B, C, and E implied that the relatively large amount of nutrients supplied from the irrigation TWW might be sufficient to meet the demands of the rice plants without the need for applying P fertilizer. However, the differences in plant height among all the treatments were not significant ($p > 0.05$). The same trend was observed in terms of the SPAD values, in which the chlorophyll meter readings were higher in the BI treatments than in the control and TI treatments. Since the chlorophyll meter reading is highly correlated with the leaf N concentration and chlorophyll content [22], it is one of the best indicators of photosynthetic activity in rice. Thus, the recorded SPAD values likely suggest that N assimilation is more effective in BI than in TI and the control irrigation method. These results might explain the high TN concentrations in the irrigation TWW and the higher N removal efficiency of Treatments A, B, C, and E compared with that of Treatments D and F. However, these variations were not statistically significant ($p > 0.05$).

**Table 4.** Rice growth, rice yield, yield components, and rice plant dry mass.

| Treatments | Plant Height (cm) | SPAD | Shoot Dry Mass (t ha⁻¹) | Panicles per Plant | Spikelets per Panicle | 1000-Grain Weight (g) | Filled Grain Percentage (%) | Rice Yield (t ha⁻¹) |
|---|---|---|---|---|---|---|---|---|
| A | 101.7 | 46.3 | 12.2 [ab] | 23.8 [ab] | 90.8 [a] | 29.2 | 90.3 | 14.1 [a] |
| B | 104.7 | 46.3 | 12.1 [ab] | 25.0 [b] | 77.8 [ab] | 28.3 | 89.4 | 12.3 [ab] |
| C | 105.7 | 47.4 | 12.9 [b] | 24.8 [b] | 79.8 [ab] | 28.7 | 89.6 | 12.5 [ab] |
| D | 99.7 | 46.0 | 8.4 [a] | 20.8 [ab] | 63.6 [b] | 30.4 | 91.3 | 9.0 [b] |
| E | 104.5 | 47.2 | 11.8 [ab] | 24.8 [b] | 80.7 [ab] | 28.8 | 90.1 | 12.8 [ab] |
| F | 99.4 | 44.9 | 9.1 [ab] | 19.5 [a] | 80.5 [ab] | 29.3 | 90.9 | 10.3 [ab] |

Values in same columns that do not contain same letters are different at significance level of 0.05.

The influence of different growing conditions was only statistically significant in the case of shoot dry mass ($p < 0.05$). The shoot dry mass reached a maximum value of 12.9 t ha$^{-1}$ in Treatment C, followed by that observed in other treatments irrigated by TWW, while the lowest value of 8.4 t ha$^{-1}$ was recorded in the control. The average shoot dry mass in the BI treatments (Treatments A, B, C, and E) was approximately 47% and 35% greater than that of the control (Treatment D) and the TI treatment (Treatment F), respectively. Interestingly, among the treatments irrigated with TWW, there was no significant difference between the treatments with (Treatment E) and without P-fertilizer application (Treatments A, B, C, and F). This indicates that rice plants might receive sufficient fertilization from TWW for effective biomass production without necessitating the supplementation of synthetic fertilizers.

### 3.3. Yield Components and Brown Rice Yield

The influence of different water and fertilization managements was significant for the brown rice yield and the majority of the yield components except for the 1000-grain weight and filled grain percentage (Table 4). There was an apparent difference in the mean values of panicles per plant between the treatments subjected to BI (Treatments A, B, C, and E) and in those subjected to the TI (Treatment F) and control (Treatment D). The number of spikelets per panicle was also significantly influenced by the irrigation methods. The maximum value of 90.8 spikelets per panicle was recorded in Treatment A, in which BI was applied, while the lowest value of 63.6 spikelets per panicle was seen in the control treatment. Similarly, the highest yield was recorded as 14.1 t ha$^{-1}$ in Treatment A, whereas it was reduced by 36.4% and 27.3% in the control (Treatment D) and TI (Treatment F), respectively.

The brown rice yield in the BI treatments (Treatments A, B, C, and E) was significantly higher than that in the TI treatment (Treatment F). This is likely because BI supplied more nutrients directly to the rice root systems, since the irrigation TWW needed to infiltrate the paddy soil layers before being discharged into the effluents, while TI allowed the irrigation TWW to flow over the soil surface and to be discharged into the effluent immediately.

There was no apparent difference in the brown rice yield between Treatment E with P-fertilizers and most of the other treatments with TWW irrigation, except for Treatment A. This confirmed that the rice plants irrigated with TWW could likely produce a high rice yield without the application of any exogenous chemical fertilizers. This was probably owing to the effective use of P in the TWW and the remaining P in the soil from the previous seasons [6]. [23] reported that rice plants could effectively uptake the nutrient content remaining in a long-term unfertilized paddy field and still maintain a yield as high as 80% of that obtained from traditionally fertilized paddy fields. However, a decrease in the concentration of P may occur in soils not supplied with P fertilizers. Thus, a long-term assessment is necessary to evaluate the changes in P content in unfertilized soil and the stability of the rice yield.

The high rice yield obtained from BI was mainly attributed to the improvement in the number of panicles per plant and spikelets per panicle. Compared with the control and TI, there was no statistically significant variation in the other yield components—namely, the filled grain percentage and 1000-grain weight. The filled grain percentage ranged from 89.4% to 91.3% in the six treatments with no distinct difference among them. The same trend was observed in the 1000-grain weight, which is consistent with the results of [24], who reported that irrigation with TWW did not increase the weight of rice grain.

### 3.4. Quality of Brown Rice

The main nutritional compositions of the harvested rice are shown in Table 5, along with the common values of the nutritional components of rice for animal feeding in Japan. The nutritional quality of feed rice strongly depends on the concentration of protein, the second major nutritional component of grains after starch. The protein levels in the current study ranged from 13.2% to 14.6% for the treatments irrigated with TWW—remarkably higher than that from the tap water irrigation (10.5%). The application of P fertilizer did not significantly increase the rice protein content. The protein content in Treatments A, B, C, E, and F were 13.5%, 14.6%, 14.5%, 14.3%, and 13.2%, respectively. No notable

difference in the rice protein content was found between Treatment A and Treatment F, implying that irrigation directions did not significantly affect rice grain protein.

The protein content in rice grain is strongly influenced by the rate of N application in the field. Similar to grain yield and whole-plant dry mass, the quality of brown rice (as estimated via protein content measurement) in the treatments that received TWW was much higher than that in the control with tap water irrigation. The highest protein content observed in the present study was higher than the highest value reported from the system having continuous irrigation with TWW in the previous season (13.1%) [6] and much greater than the highest value from circulated irrigation (Watanabe et al. 2016a). This is because the amount of N supplied to the rice field increased from 6.7 g (in 220 L of irrigation TWW) in circulated irrigation to 18.2 g in this continuous irrigation (596 L of irrigation TWW). Additionally, the protein content reported in this study was far higher than that recommended by the Japanese standard of feed compositions (2009) (8.8%), as well as the grain protein content of the same variety (Bekoaoba) cultivated in Japan [25] (6.2–7.0%). Protein is a key factor that influences the eating quality of rice. A high protein content may reduce the eating quality of rice for human consumption, but this rice is preferable for use as animal feed. Hence, the high levels of protein in rice reported in this study are preferable in husbandry farming.

Rice crude fat is a good source of linoleic acid and other vital fatty acids, but it does not contain cholesterol [26]. The fat content in this study ranged from 2.3% to 2.7%, similar to the results of a previous study [27] but slightly lower than the value in the standard. These results show that rice fat content was not significantly influenced by TWW irrigation, irrigation direction, or fertilizer application.

The presence of fiber in food increases the bulk of feces, improves bowel function, and helps prevent digestive disorders [28]. The crude fiber observed in the rice produced in the present study varied from 0.40% to 0.67%, which was slightly lower than the common values (Table 5) but comparable with that observed in the rice produced in an earlier study [27]. Table 5 indicates that the fiber content was not significantly influenced by TWW irrigation, irrigation direction, or fertilization. For humans, fiber-rich food helps to improve proper bowel function and diminishes the risk of developing intestinal disorders. However, low-fiber food may promote the fattening of animals. For many animals such as the cow and horse, forages other than rice are the main source of dietary fiber. Therefore, a slight difference in the fiber content of rice and feed standards is negligible.

**Table 5.** Grain quality (%) of brown rice.

| Treatments | Protein | Fat | NFE | Fiber | Ash |
|------------|---------|-----|-----|-------|-----|
| A | 13.5 [b] | 2.5 | 81.9 [a] | 0.61 | 1.7 [ab] |
| B | 14.6 [b] | 2.7 | 80.9 [a] | 0.40 | 1.8 [b] |
| C | 14.5 [b] | 2.3 | 81.4 [a] | 0.59 | 1.7 [b] |
| D | 10.5 [a] | 2.7 | 84.7 [b] | 0.59 | 1.8 [b] |
| E | 14.3 [b] | 2.6 | 81.1 [a] | 0.67 | 1.7 [ab] |
| F | 13.2 [b] | 2.6 | 82.2 [a] | 0.62 | 1.6 [a] |
| Common values * | 8.8 | 3.2 | 85.6 | 0.8 | 1.6 |

*: Adapted from [29]; values in same columns that do not contain same letters different at significance level of 0.05.

The nitrogen-free extract (NFE) in this work varied from 80.9% to 84.7%, which is lower than the corresponding value generally reported in feed compositions in Japan. This is likely due to the higher temperature in the apparatus after the heading stage. Similar to a report of [27], the transparent roof that was used to avoid the effect of rainfall tended to trap the heat and increased the temperature in the apparatus zone, which was the primary reason for the reduction in NFE. NFE was not affected by the irrigation direction but was significantly decreased by TWW irrigation compared with that of tap water irrigation.

Ash content represents the total mineral content in foods. The values for the percentage of ash content obtained in the current study ranged between 1.6% and 1.8%, which is slightly higher than the

common value (1.6%) as shown in Table 5. All of these values in all treatments were slightly lower than the common values, and these values did not significantly vary between treatments. The negative correlation between protein and other nutritional components suggests that rice cultivars high in protein may be low in other nutrients. These results are well supported by the findings of [30].

### 3.5. Accumulation of Heavy Metals in Paddy Soil and Brown Rice

Alghobar and Suresha [24] reported that TWW irrigation, compared with well water irrigation, significantly increased heavy metals such as Mn, Cu, Cd, and Ni in the paddy soil. A slight increase in Cd, Cu, Pb, and Zn in soil was observed in domestic wastewater irrigation compared to groundwater irrigation [31]. The adsorption of heavy metals in soil is influenced by many factors such as soil pH, OM concentration. [32] reported that increase in OM markedly increased adsorption of Pb, Cu, Cd, Zn in soil. It is well-known that harmful metals in agricultural soils may affect both the crop yield and quality [33]. For this experiment, the concentrations of heavy metals in the paddy soils are listed in Table 6. Relative to the initial soils, heavy metal concentrations in the soils after the experiment showed no considerable difference among the treatments that received TWW. However, a slight increase in Cu content was observed in the soil of the control that was irrigated with tap water. This was attributed to the oxidation of a section of copper cable used for the MFC system. Treatments B, C, and E repeatedly used the soil from the previous season [6], in which Cu concentrations were much higher than in the new soil used for the Treatments A, D, and F.

The mean concentrations of Fe, Mn, Cu, Zn, Ni, Pb, Cr, and As in the harvested brown rice from all treatments are presented in Table 7. The irrigation with TWW did not significantly influence the buildup of heavy metals in rice grain. The significant difference between the Cu concentration in the rice harvested from Treatments A and B was probably due to the notable difference in Cu content in the initial soils of both treatments.

**Table 6.** Heavy metal contents (mg kg$^{-1}$) in soils before (initial) and after experiment (after harvesting).

|  | New Soil | | | | Old Soil | | | |
| --- | --- | --- | --- | --- | --- | --- | --- | --- |
|  | Initial | After Harvesting | | | Initial | After Harvesting | | |
|  | | Treatment A | Treatment D | Treatment F | | Treatment B | Treatment C | Treatment E |
| Fe | 53.9 | 48.2 | 53.6 | 53.3 | 70.8 | 56.2 | 51.6 | 52.8 |
| Mn | 380.3 | 300.8 | 350.7 | 343.6 | 392.1 | 389.8 | 391.2 | 352.8 |
| Cu | 17.0 | 15.0 | 45.1 | 22.4 | 97.1 | 113.4 | 98.7 | 94.3 |
| Zn | 98.0 | 86.9 | 103.3 | 101.3 | 119.2 | 127.4 | 120.1 | 117.0 |
| Cd | 0.06 | 0.07 | 0.05 | 0.04 | 0.07 | 0.05 | 0.05 | 0.04 |
| Ni | 20.3 | 18.8 | 20.3 | 20.3 | 21.5 | 24.1 | 23.1 | 21.6 |
| Pb | 12.8 | 11.4 | 13.3 | 13.4 | 16.1 | 16.6 | 16.9 | 15.7 |
| Cr | 20.5 | 18.2 | 21.9 | 22.7 | 28.5 | 32.0 | 30.4 | 28.7 |
| As | 1.5 | 5.6 | 2.0 | 2.0 | 3.3 | 2.9 | 1.7 | 1.3 |

**Table 7.** Concentrations (mg kg$^{-1}$) of heavy metals in brown rice.

| Treatments | Fe | Mn | Cu | Zn | Cd | Ni | Pb | Cr | As |
| --- | --- | --- | --- | --- | --- | --- | --- | --- | --- |
| A | 16.9 | 29.1 | 4.9 [a] | 15.0 | 0.03 [a] | 0.31 [ab] | 0.03 | 0.04 | 0.16 [ab] |
| B | 14.2 | 30.1 | 7.4 [c] | 15.1 | 0.04 [a] | 0.49 [b] | 0.03 | 0.05 | 0.13 [a] |
| C | 13.3 | 26.3 | 6.3 [abc] | 14.1 | 0.03 [a] | 0.36 [ab] | 0.03 | 0.04 | 0.17 [b] |
| D | 11.9 | 26.4 | 5.2 [ab] | 14.2 | 0.04 [a] | 0.23 [a] | 0.02 | 0.05 | 0.25 [c] |
| E | 13.0 | 29.9 | 6.6 [abc] | 14.8 | 0.04 [a] | 0.46 [b] | 0.03 | 0.04 | 0.16 [ab] |
| F | 12.3 | 33.4 | 6.9 [bc] | 14.8 | 0.09 [b] | 0.68 [c] | 0.02 | 0.03 | 0.14 [a] |
| Permissible Limits * | NA | NA | NA | NA | 0.40 | NA | 0.20 | NA | NA |

*: Adapted from [34]; values in same columns that do not contain same letters are different at significance level of 0.05.

Cd and Pb are nonessential elements that may be phytotoxic to sensitive species at low concentrations [31]. The levels of Cd, Cu, and Zn in the rice from this study varied from 0.03 to 0.09, 4.9 to 7.4, and 14.1 to 15.1 mg kg$^{-1}$, respectively—lower than those in the most common variety of rice in Japan [35]. The concentrations of Cd and Pb in the brown rice from all treatments were well below the safe limits set by the FAO/WHO (2004) and EU communities (2006) [34,36]. The content of heavy metals in brown rice during this season was comparable to that obtained in a previous study [6]. However, the continuous monitoring of these harmful metals in brown rice and soil is necessary to avoid potential long-term accumulation or accidental contamination when the same paddy fields are repeatedly used for rice production with TWW irrigation. The overall mass balances of heavy metals in the experiment, which were calculated by multiplying the dry weight of rice or soil (or the volume of irrigation water) by the concentration of heavy metals in rice, soil, and water, are illustrated in Table 8. Most of the masses of the heavy metals accumulated in whole rice plants were comparable with the masses of those metals in the influent water. However, amount of Mn accumulated in the whole rice plants was much greater than that from influent water, while only 10% of Ni from influent water was accumulated in rice plant.

**Table 8.** Overall mass balances (mg) of heavy metals in the experiment (treatments A, B, C, E, and F).

| Sources | Fe | Mn | Cu | Zn | Cd | Ni | Pb | Cr | As |
|---|---|---|---|---|---|---|---|---|---|
| **Input** | | | | | | | | | |
| Influent water | 239.4 | 85.4 | 34.3 | 109.4 | NA | 28.1 | 1.6 | 1.6 | NA |
| Soil before experiment | 6371.2 | 38,506.7 | 6497.0 | 11,005.7 | 6.4 | 2088.9 | 1472.0 | 2515.2 | 259.2 |
| Total | 6610.6 | 38,592.1 | 6531.3 | 11,115.1 | NA | 2117.0 | 1473.6 | 2516.7 | NA |
| **Output** | | | | | | | | | |
| Effluent water | 326.2 | 68.2 | 254.7 | 70.0 | NA | 13.7 | 14.2 | 1.8 | NA |
| Soil after experiment | 5211.1 | 35,353.5 | 7290.5 | 11,029.5 | 5.4 | 2147.5 | 1467.4 | 2609.6 | 270.1 |
| Rice * | 282.7 | 2973.5 | 46.3 | 99.7 | 0.5 | 2.9 | 0.9 | 1.3 | 0.2 |
| Total | 5820.0 | 38,395.2 | 7591.5 | 11,199.2 | 5.8 | 2164.1 | 1482.5 | 2612.7 | NA |

*: The whole rice plant including rice seed, leaf, stem, and root. NA: Not available.

### 3.6. Necessity of Fertilizer Application

The nutrient inputs in the paddy fields were calculated by multiplying the concentrations of N, K, and P in TWW (Table 2) by the amount of TWW used for irrigation in each treatment. This indicated that the amounts of N and K input in the TWW irrigation treatments were considered sufficient as compared with that in the control (Table 9). By contrast, the amount of P supplied from TWW in Treatments A, B, C, and F was only 6 kg ha$^{-1}$, which was much less than that from the supplementation of the synthetic P fertilizer in Treatment D (160 kg $P_2O_5$ ha$^{-1}$). Therefore, we examined the change in P content of the soils before and after the experiment (Table 3). While other treatments irrigated with TWW exhibited a decrease in P content in the soil, an increase in P concentration in the soil of Treatment E was found. This may be attributed to the application of the P fertilizer. An increased level of P was also observed in Treatment D, which is likely associated with less plant growth and a lower yield of rice in this treatment than in the other treatments (Table 4). The decrease in P in soil under TWW irrigation without fertilization could make the soil less fertile. It is obvious that both P concentration in TWW and the rate of TWW irrigation contribute to the P balance of the system and determine whether or not P fertilization is necessary [9]. In this study, higher rice yield and quality could be obtained in one season, but for long-term use, P content in the experimental soil needs to be assessed in future studies.

**Table 9.** Total nutrient (kg ha$^{-1}$) supply to paddy field from TWW and fertilizers.

| Treatment | N | K | P |
|-----------|------|-----|-----|
| A | 1013 | 428 | 6 |
| B | 1013 | 428 | 6 |
| C | 1013 | 428 | 6 |
| D | 260 | 260 | 160 |
| E | 1013 | 428 | 166 |
| F | 1013 | 428 | 6 |

### 3.7. Electric Output from MFC System

Figure 4a shows the variation in electricity generated from the MFC systems. Before the heading stage, the electric output obtained from the MFC system was lower than 50 mV, which is equivalent to 0.14 mW m$^{-2}$. This was much lower than the observations in normal paddy fields [11]. During MSD, the MFC system nearly stopped since the soil was kept dry. After that, since the paddy fields were re-flooded, the electric output increased rapidly and reached 4.2 mW m$^{-2}$ in Treatment A and 2.8 mW m$^{-2}$ in Treatment E. The lower output from Treatments D and F may support our hypothesis that more electricity can be gained by supplying more OM from TWW.

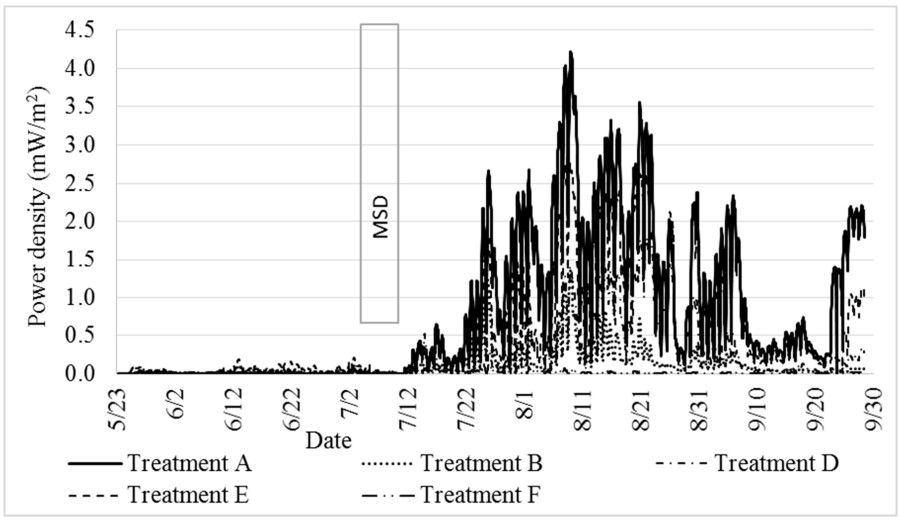

(a)

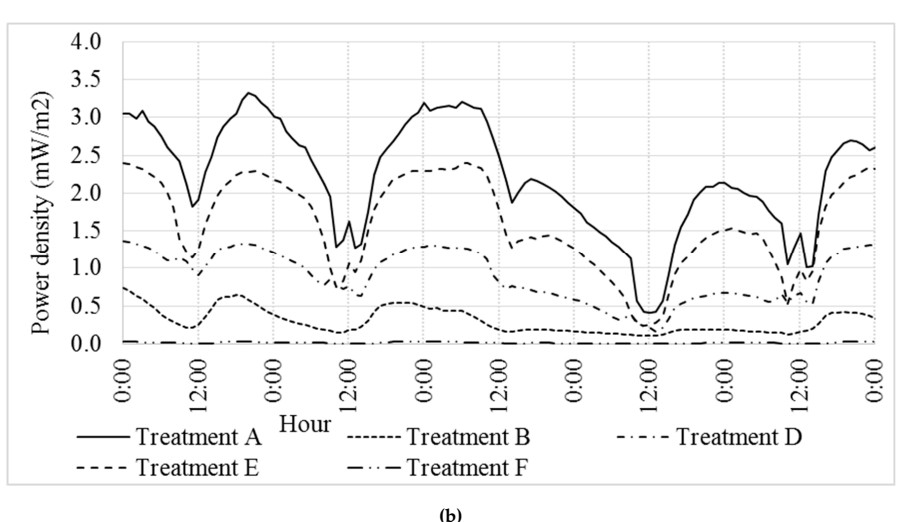

(b)

**Figure 4.** Power densities from the MFC systems throughout cultivation period (**a**) and only in five days of ripening stage (**b**). MSD means midsummer drainage to dry up soil layers.

Figure 4b shows the day/night cycles of the output fluctuations during a five-day-period in August (flowering stage). Surprisingly, the electric output decreased during the day and increased at night, demonstrating an opposite trend to those observed in previous studies using plant-type MFCs [11,13,14,17,37]. Photosynthesis causes OM creation via root exudates in the anodic compartment, which has a positive effect on the MFC performance. However, photosynthesis also generates oxygen through the rice roots, which increases the redox potential in the root zone and thus has a negative effect on electricity generation. The results obtained in the present work indicated that sunlight caused a decrease in electricity output. Thus, it is clear that the oxygen production effect outweighed the exudate generation.

## 4. Conclusions

The results obtained in this study show that the TWW used for irrigation satisfied the irrigation water quality criteria and that there were no adverse effects of TWW irrigation (Treatment A, B, C, E, and F) on the characteristics of rice growth, yield, and yield components, or on the heavy metal content in soils and brown rice, compared with those in the control (Treatment D). In addition, there were positive effects of TWW irrigation on rice growth and yield capacity compared with conventional irrigation. Overall, the safety and suitability of TWW irrigation in rice cultivation were demonstrated in this present study with the following highlights:

- A high yield and nutritional quality of brown rice could be achieved by continuous TWW irrigation without any fertilizer application, especially in BI, which was superior to both TI and the conventional irrigation method. Nevertheless, monitoring P in the soil after each season is still recommended to determine whether the application of P fertilizers is required in subsequent growing seasons.
- No hazardous metals accumulated in the soil or in the harvested rice when the soil was used repeatedly with continuous TWW irrigation. However, monitoring heavy metals in the soil and brown rice every season is highly recommended to fully characterize the effects of TWW irrigation.

The electric output from the MFC systems was still low compared with that reported from normal paddy fields, even when the connection was modified using graphite rods instead of copper cables. Further studies are necessary to optimize the potential of MFC systems in paddy fields.

**Author Contributions:** D.D.P.: Design and conduct the experiment in general, analyze water and rice samples, and write manuscript. K.C.: Conduct the experiment. L.D.P.: Conduct the statistical data analysis and write manuscript. N.K.: Make advices on MFC system. A.S.: Analyze metals in rice and soil samples. Y.S.: Make advices on rice cultivation. K.H.: Analyze nutrition in rice samples. D.V.P.: Conduct the experiment. T.W.: Generally supervise for the experiment, check and revise manuscript.

**Funding:** This study was supported by the Ministry of Land, Infrastructure, Transport, and Tourism (MLIT) of Japan through the Gesuido Academic Incubation to Advanced (GAIA) project, and by the Ministry of Education, Sports, Culture, Science, and Technology (MEXT) of Japan through the Center of Community (COC) project. This study was conducted as an activity of the Institute for Regional Innovation, Yamagata University.

**Acknowledgments:** We sincerely appreciate Division of Sewerage in Tsuruoka City Government and Tsuruoka Branch of Japan Agricultural Cooperatives (JA) for their strong supports as collaborators in this study.

**Conflicts of Interest:** The authors declare no conflict of interest.

## Abbreviations

EC: Electrical conductivity; K: Potassium; MSD: Midseason drainage; NFE: Nitrogen-free extract; SPAD: Soil plant analysis development; TN: Total nitrogen; TOC: Total organic carbon; TP: Total phosphorus; TWW: Treated wastewater; WWTP: Wastewater treatment plant

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
