# Peer review of "Rice Cultivation without Synthetic Fertilizers and Performance of Microbial Fuel Cells (MFCs) under Continuous Irrigation with Treated Wastewater"

_water, doi:10.3390/w11071516_

Reviewer 1 Report

The response of rice plants and distribution of nutrients and heavy metals were tested in a small scale experiment including containers receiving either treated municipal wastewater or synthetic fertilizers. Influent was applied with two different ways from bottom to top and top to top. The study generally address the research questions, however revisions are needed to improve the quality of the manuscript and help the potential reader.

Specific comments:

L92. Properties of the soil(s) used in this study should be presented in text or table including probably PH EC nutrient organic matter and heavy metals concentrations.

L113-116. It would be better to explain further fertilization. For example kg/ha refer to the area of the soil and nutrient concentrations?

L117 Table 1 should be explained further in the materials and methods section to help the reader. Experimental design would be ideal if the treatments (A, B,…) could cover similar conditions. For example BI was tested in with fertilization,   in new-old, and P addition, but TI only without fertilization in new soil. This would give some more “area” for more robust statistical comparisons between BI and TI.   The reason of not using this experimental design at least should be included in the materials and methods section.

L179. N removal efficiency should be explained further and showed probably in figure which could include time in x-axis and concentrations of influent and effluent (and %) in two y-axes.  

L153. I would suggest to transform totally table 3 to a table showing the amount of nutrients nutrients added to the system (table 8) and subsequent distributions to soil, plant (uptake) and effluent indicating in parallel losses to the atmosphere (case of N). In such a case reader would have available a complete overview of the fate of nutrients across the system examined.  

L246, 277 Change table 4 to 5.

L299. It would be valuable to see also here the overall mass balances of heavy metals including also distribution percentages across soil-plant-effluent system.

Author Response

1.         L92. Properties of the soil(s) used in this study should be presented in text or table including probably PH EC nutrient organic matter and heavy metals concentrations.

Response: It is a pity that we did not measured pH, EC and OM of the soil used for the experiment. For nutrients and heavy metals concentrations, we measured it and those parameters are presented in Table 3 and Table 6.

2.         L113-116. It would be better to explain further fertilization. For example kg/ha refer to the area of the soil and nutrient concentrations?

Response: The fertilization has been explained more detailed as follows: Fertilization was accomplished by applying a commercial N-P-K fertilizer (20.6 g N-P-K fertilizer with 14 % N, 14 % P, 14 % K for 0.18 m2 equivalent to 160 kg N ha-1, 160 kg P ha-1, and 160 kg K ha-1) before transplanting and an N-K fertilizer (9.0 g N-K fertilizer with 14 % N and 14 % K for 0.18 m2 equivalent to 100 kg N ha-1 and 100 kg K ha-1) as a top dressing at the panicle initiation stage (FNPK).

3.         L117 Table 1 should be explained further in the materials and methods section to help the reader. Experimental design would be ideal if the treatments (A, B,…) could cover similar conditions. For example BI was tested in with fertilization,   in new-old, and P addition, but TI only without fertilization in new soil. This would give some more “area” for more robust statistical comparisons between BI and TI.   The reason of not using this experimental design at least should be included in the materials and methods section.

Response: It has been edited.

4.         L179. N removal efficiency should be explained further and showed probably in figure which could include time in x-axis and concentrations of influent and effluent (and %) in two y-axes.  

Response: Concentrations of TN in the influent and effluent tanks has been added in Figure 2.

5.         L153. I would suggest to transform totally table 3 to a table showing the amount of nutrients added to the system (table 8) and subsequent distributions to soil, plant (uptake) and effluent indicating in parallel losses to the atmosphere (case of N). In such a case reader would have available a complete overview of the fate of nutrients across the system examined.  

Response: We would like to keep the format of those tables.

6.         L246, 277 Change table 4 to 5.

Response: It has been revised.

7.         L299. It would be valuable to see also here the overall mass balances of heavy metals including also distribution percentages across soil-plant-effluent system.

Response: Overall mass balances of heavy metals has been added in Table 8.

Reviewer 2 Report

Review for the article titled “Rice cultivation without synthetic fertilizers and performance of microbial fuel cells (MFCs) under continuous irrigation with treated wastewater”

General comment

This is a well written article combining key sustainability concepts (renewable energy, recycling of nutrients and water) of great importance in the current time. I recommend publication with minor revisions. This would be an interesting read for the research communities. I hope that primary authors can include treatment replications, and move from batch scale to field scale in their future work.

Table 1 is difficult to read. I will suggest to make it legible by using vertical lines to show and separate each treatment. Also, some undefined terms (see specific comment below) have been used inside the table, which should be avoided.

Key: [x, y] = line number x on page y of 14

[2:66] It will be useful to mention the studies (e.g. Jaiswal and Elliott, 2011) showing the long-term impact of TWW irrigation on soil P fertility by adding a sentence at the end of the paragraph ending with “....without necessitating the supplementations of synthetic fertilizers”.

A possible example sentence could be:

“However, rate of soil P buildup may slow down and eventually reach steady-state after long-term TWW irrigation (Jaiswal and Elliott, 2011) due to downward movement of P binding minerals. In such case, P fertilization might again become necessary if P input with TWW and steady-state soil P levels are insufficient to meet the P demand.”

[2:76] mention essential plant nutrients (N, P, K). More (micronutrients),  if the information is available based on the composition of TWW.

[2:89] Different conditions of continuous irrigation is ambiguous. Please be more specific about what are those conditions.

[3: 98] The labels (a and b) are missing from Figure 1.

[3:101] Were pumps not used for BI design? I appears (from Figure 1) then pumps were used. If so, please mention is here clearly. Now, from the way the sentence is written, it appears that pumps were used only in TI system and not in BI system.

[3:113] what is symbol FNPK?

[3:117]  an explanation for closed and opened MFC circuit is needed.

[5: 166] what do you mean by “four hills of rice”. I am not familiar with this term (“hills). It will be useful to provide and alternative term.

[8:294] When comparing results to standard, reference to the standard and standard values are missing. This needs to be corrected; e.g. ash content at .

[8:299] In  discussing accumulation of heavy metals in soils, Some discussion is required about the long-term fate of heavy metals accumulations in the context of competitive adsorption and soil organic matter, and pH (Elliott et al. 1986).

[9:332] In discussing the necessity of fertilizer application It is required to mention that TWW may come from different sources (municipal, industry, etc.) and they have different  P concentration. Both the P concentration, and rate of TWW irrigation (Elliott and Jaiswal, 2011) contribute to P balance of the system, and determines whether P fertilization is necessary or not.

references

Jaiswal, D., and H. A. Elliott. "Long-term phosphorus fertility in wastewater-irrigated cropland." Journal of environmental quality 40, no. 1 (2011): 214-223.

Elliott, H. A., and D. Jaiswal. "Phosphorus management for sustainable agricultural irrigation of reclaimed water." Journal of Environmental Engineering 138, no. 3 (2011): 367-374.

Elliott, Herschel Adams, M. R. Liberati, and C. P. Huang. "Competitive Adsorption of Heavy Metals by Soils 1." Journal of Environmental Quality 15, no. 3 (1986): 214-219.

Author Response

1.      [2:66] It will be useful to mention the studies (e.g. Jaiswal and Elliott, 2011) showing the long-term impact of TWW irrigation on soil P fertility by adding a sentence at the end of the paragraph ending with “....without necessitating the supplementations of synthetic fertilizers”. A possible example sentence could be: “However, rate of soil P buildup may slow down and eventually reach steady-state after long-term TWW irrigation (Jaiswal and Elliott, 2011) due to downward movement of P binding minerals. In such case, P fertilization might again become necessary if P input with TWW and steady-state soil P levels are insufficient to meet the P demand.”

Response: The sentences have been added to the manuscript.

2.      [2:76] mention essential plant nutrients (N, P, K). More (micronutrients),  if the information is available based on the composition of TWW.

Response: Some micronutrients have been added.

3.      [2:89] Different conditions of continuous irrigation is ambiguous. Please be more specific about what are those conditions.

Response: We will consider it in the further study.

4.      [3: 98] The labels (a and b) are missing from Figure 1.

Response: It has been added in the manuscript.

5.      [3:101] Were pumps not used for BI design? I appears (from Figure 1) then pumps were used. If so, please mention is here clearly. Now, from the way the sentence is written, it appears that pumps were used only in TI system and not in BI system.

Response: It has been revised to explain that pumps were used for both BI and TI.

6.      [3:113] what is symbol FNPK?

Response: FNPK means the fertilizer regime that N-P-K fertilizer was used.

7.      [3:117]  an explanation for closed and opened MFC circuit is needed.

Response: More explanation has been added.

8.      [5: 166] what do you mean by “four hills of rice”. I am not familiar with this term (“hills). It will be useful to provide and alternative term.

Response: One hill of rice normally includes 3-5 young rice seedlings.

9.      [8:294] When comparing results to standard, reference to the standard and standard values are missing. This needs to be corrected; e.g. ash content at .

Response: Standard and value have been added.

10. [8:299] In  discussing accumulation of heavy metals in soils, Some discussion is required about the long-term fate of heavy metals accumulations in the context of competitive adsorption and soil organic matter, and pH (Elliott et al. 1986).

Response: More discussion has been added.

11. [9:332] In discussing the necessity of fertilizer application It is required to mention that TWW may come from different sources (municipal, industry, etc.) and they have different  P concentration. Both the P concentration, and rate of TWW irrigation (Elliott and Jaiswal, 2011) contribute to P balance of the system, and determines whether P fertilization is necessary or not.

Response: It has been added
